# Penalized ensemble Kalman filters for high dimensional non-linear systems

**Elizabeth Hou**[1]*, **Earl Lawrence**[2], **Alfred O. Hero**[1]

**1** EECS Department, University of Michigan, Ann Arbor, Michigan, United States of America, **2** Los Alamos National Laboratory, Los Alamos, New Mexico, United States of America

* emhou@umich.edu

**Data Availability Statement:** All relevant data and simulation code are uploaded to GitHub and accessible via the following URL: https://github.com/robertsy/assimilr.

## Abstract

The ensemble Kalman filter (EnKF) is a data assimilation technique that uses an ensemble of models, updated with data, to track the time evolution of a usually non-linear system. It does so by using an empirical approximation to the well-known Kalman filter. However, its performance can suffer when the ensemble size is smaller than the state space, as is often necessary for computationally burdensome models. This scenario means that the empirical estimate of the state covariance is not full rank and possibly quite noisy. To solve this problem in this high dimensional regime, we propose a computationally fast and easy to implement algorithm called the penalized ensemble Kalman filter (PEnKF). Under certain conditions, it can be theoretically proven that the PEnKF will be accurate (the estimation error will converge to zero) despite having fewer ensemble members than state dimensions. Further, as contrasted to localization methods, the proposed approach learns the covariance structure associated with the dynamical system. These theoretical results are supported with simulations of several non-linear and high dimensional systems.

## 1 Introduction

The Kalman filter is a well-known technique to track a linear system over time, and many variants based on the extended and ensemble Kalman filters have been proposed to deal with non-linear systems. The ensemble Kalman filter (EnKF) [1, 2] is particularly popular when the non-linear system is extremely complicated and its gradient is infeasible to calculate, which is often the case in geophysical systems. However, these systems are often high dimensional and forecasting each ensemble member forward through the system is computationally expensive. Thus, the filtering often operates in the high dimensional regime where the number of ensemble members, $n$, is much less than the size of the state, $p$. It is well known that even when $p/n \to const.$ and the samples are from a Gaussian distribution, the eigenvalues and the eigenvectors of the sample covariance matrix do not converge to their population equivalents, [3, 4]. Since our ensemble is both non-Gaussian and high dimensional ($n \ll p$), the sample covariance matrix of the forecast ensemble will be extremely noisy. In this paper, we propose a variant of the EnKF specifically designed to handle covariance estimation in this difficult regime,

**Funding:** This work was supported by the Consortium for Verification Technology under Department of Energy National Nuclear Security Administration award numbers DE-NA0002534 and DE-NA0003921 to AOH and DE-NA000253 to EH, and the Laboratory Directed Research and Development program at Los Alamos National Laboratory under project 20150033DR, SHIELDS: Space Hazards Induced near Earth by Large Dynamic Storms - Understanding, Modeling, and Predicting to EL.

**Competing interests:** The authors have declared that no competing interests exist.

but with weaker assumptions and requiring less prior information on the state covariance structure than competing approaches.

## 1.1 Related work

To deal with the sampling errors, many schemes have been developed to de-noise the forecast sample covariance matrix. These schemes "tune" the matrix with variance inflation and localization, [5–18]. However, these schemes are often not trivial to implement because they require carefully choosing the inflation factor and using expert knowledge of the true system to set up the localization. Additionally, the EnKF with perturbed observations introduces additional sampling errors due to the perturbation noise's lack of orthogonality with the ensemble. Methods have been devised that construct perturbation matrices that are orthogonal, [19]; however these methods are computationally expensive, [20]. This has led to the development of matrix factorization versions of the EnKF such as the square root and transform filters, [20–27], which do not perturb the observations and are designed to avoid these additional sampling errors. Recent work [28] has also revisited the stochastic EnKF, which perturbs the modeled observations instead of the observations themselves arguing that in regimes with skewed likelihoods, this method is more accurate.

The ensemble Kalman filter is closely related to the particle filter [29, 30], although it uses a Gaussian approximation of the conditional state distribution in order to get an update that is a closed form expression for the analysis ensemble (as opposed to one that requires numerical integration). While the particle filter does not use this approximation, it also requires an exponential number of particles to avoid filter collapse, [31]. Recently, there has been significant effort to apply the particle filter to larger scale systems using equal weights, [32, 33], and merging it with the ensemble Kalman filter to form hybrid filters, [29, 34–37]. The EnKF is also related to the unscented Kalman filter, [38, 39], which handles nonlinearity by propagating a carefully selected set of "sigma points" (as opposed to the randomly sampled points of the EnKF) through the nonlinear forecast equations. The results are then used to reconstruct the forecasted mean and covariance.

Most similar to our proposed work are [40] and [41], which also propose methods that use sparse inverse covariance matrices. Both methods justify the appropriateness of using the inverse space with large scale simulations or real weather data. The former reports that their computational complexity is polynomial in the state dimension and requires the stronger assumptions of Gaussianity and structural knowledge. The latter algorithm can be implement in parallel making it very efficient, however, the paper [41] still makes the *much* stronger assumptions of Gaussianity and known conditional independence structure among the states.

## 1.2 Proposed method

We propose a penalized ensemble Kalman filter (PEnKF), which uses an estimator of the forecast covariance whose inverse is sparsity regularized. While the localization approaches effectively dampen or zero out entries in the covariance, *our approach zeros out entries in the inverse covariance, resulting in a sparse inverse covariance*. This provides two advantages. First, it makes a weaker assumption about the relationship between state variables. Second, our approach does not require anything like localization's detailed knowledge of which covariance entries to fix at zero or how much to dampen. Instead, it merely favors sparsity in the inverse covariance. Additionally, our method is very easy to implement because it just requires using a different estimator for the covariance matrix in the EnKF. We can explicitly show the improvement of our estimator through theoretical guarantees.

**1.2.1 Outline.**  In Section 2, we explain the assumptions in our high-dimensional system and we give background on the EnKF and $\ell_1$ penalized inverse covariance matrices. In Section 3, we give details on how to modify the EnKF to our proposed PEnKF and provide theoretical guarantees on the filter. Section 4 contains the simulation results of the classical Lorenz 96 system and a more complicated system based on modified shallow water equations.

## 2 Background

In this paper, we consider the scenario of a noisy, non-linear dynamics model $f(\cdot)$, which evolves a vector of unobserved states $\mathbf{x}_t \in \mathbb{R}^p$ through time. We observe a noisy vector $\mathbf{y}_t \in \mathbb{R}^r$, which is a transformation of $\mathbf{x}_t$ by a function $h(\cdot)$. Both the process noise $\boldsymbol{\omega}_t$ and the observation noise $\epsilon_t$ are independent of the states $\mathbf{x}_t$. We assume both noises are zero mean Gaussian distributed with known diagonal covariance matrices, $\mathsf{Q}$ and $\mathsf{R}$. Often, it is assumed that the dynamics model does not have noise, making $\boldsymbol{\omega}_t$ a zero vector, but for generality we allow $\boldsymbol{\omega}_t$ to be a random vector.

$$\mathbf{x}_t = f(\mathbf{x}_{t-1}) + \boldsymbol{\omega}_t \qquad \text{Dynamics Model}$$
$$\mathbf{y}_t = h(\mathbf{x}_t) + \epsilon_t \qquad \text{Observation Model}$$

As with localization methods, we make an assumption about the correlation structure of the state vector in order to handle the high dimensionality of the state. In particular, we assume that only a small number $s \ll \binom{p}{2}$ of pairs of state variables have non-zero conditional correlation, $\text{Cov}(x_i, x_j | x_{-(i,j)}) \neq 0$ where $x_{-(i,j)}$ represents all state variables except $x_i$ and $x_j$. This means that, given all of the rest of the state, $x_i$ and $x_j$ are conditionally uncorrelated. They may have a dependency, meaning that the marginal correlation between them is non-zero, but such dependency is entirely explained by conditional dependence on other parts of the state. An example is given by a one-dimensional spatial field with three locations $x_1$, $x_2$, and $x_3$ where $x_1$ and $x_3$ are both connected to $x_2$, but not each other. In this case, it might be reasonable to model $x_1$ and $x_3$ as uncorrelated conditioned on $x_2$ although they are not necessarily marginally uncorrelated. Their marginal correlation might not be zero, but their conditional or partial correlation is zero. Rather than assuming that the zero pattern in the matrix of conditional correlations has a specific structure, we place no assumption on the zero pattern, only on the number of zeros (sparsity) of this matrix. In other words, we will allow the data to determine the specific pattern.

The assumption that the set of non-zero conditional correlations is sparse is equivalent to the assumption that the inverse correlation matrix of the model state is sparse with few non-zero off-diagonal entries [42]. We can also quantify the sparsity level as $d$, which is the maximum number of non-zero off-diagonals in any row. Thus the sparsity assumption can be stated as $d^2 \ll p^2$. Note that our assumption is on the conditional *correlation* or lack of it, and we do not make any claims on independence between states. This is because $\mathbf{x}_t$ is not Gaussian when $f(\cdot)$ is non-linear so lack of correlation does not imply any independence. Thus the zeros in the inverse covariance matrix do not imply conditional independence. This assumption is weaker than the one commonly made in localization. This common assumption is equivalent to assuming that the marginal covariance matrix itself is sparse whereas our assumption of sparse conditional covariance admits a dense covariance. Finally, because we do not assume that the conditionally correlated state variable interactions are the same for different time points, we allow the support set $\mathcal{E}_t$ (set of non-zero conditional correlations) and its size $s_t$ to change over time.

While we assume $f(\cdot)$ can be highly non-linear, which occurs in many geophysical systems, we assume the measurement system $h(\cdot)$ is relatively linear, e.g, piece-wise linear, with gradients that are feasible to calculate. In the following subsections, we will use $\mathsf{H}$ to represent a linear operator equal to the Jacobian of $h(\cdot)$ when the measurement system is non-linear, under the assumption that $\mathsf{H}$ adequately approximates $h(\cdot)$. Additionally we assume that the additive noise $\boldsymbol{\omega}$ in the measurement system is Gaussian.

## 2.1 Ensemble Kalman filter

The EnKF of [19] is a well studied algorithm and there are numerous improved models [20–27] that build on its foundation. At time $t = 0$, $n$ samples are drawn from an initial distribution, which is often chosen as the standard multivariate normal when the true initial distribution is unknown, to form an initial ensemble $\mathsf{A} \in \mathbb{R}^{p \times n}$. Subsequently, at every time point $t$, the observations $\mathbf{y}_t$ are perturbed $n$ times with Gaussian white noise, $\eta^j \sim N(\mathbf{0}, \mathsf{R})$, to form a perturbed observation matrix $\mathsf{D}_t \in \mathbb{R}^{p \times n}$, where $\mathbf{d}_t^j = \mathbf{y}_t + \eta^j$.

The forecast covariance estimator $\hat{\mathsf{P}}^f$ is often defined as the sample covariance of the forecast ensemble, equal to $\mathsf{S} = \frac{1}{n-1}(\mathsf{A}_0 - \bar{\mathsf{A}}_0)(\mathsf{A}_0 - \bar{\mathsf{A}}_0)^T$, where $\bar{\mathsf{A}}_0$ is a $p \times n$ matrix whose columns are the sample mean vector $\frac{1}{n}\sum_{j=1}^n \mathbf{a}_0^j$, but it can be another estimator such as a localized estimator (one that is localized with a taper matrix), or a penalized estimator as proposed in this paper.

## 2.2 Bregman divergence and the $\ell_1$ penalty

Below, we give a brief overview of the $\ell_1$-penalized log-determinant Bregman divergence and some properties of its minimizer, as described in [43]. We denote $\mathsf{S}$ to be any arbitrary sample covariance matrix, and $\Sigma = \mathrm{E}(\mathsf{S})$ to be its true covariance matrix, where $\mathrm{E}(\cdot)$ is the expectation function.

Bregman divergence $\mathrm{B}(\Theta\|\Omega)$ between two functions $\Theta$ and $\Omega$ is a measure of difference between the two functions. Here the functions to be compared are covariance matrices. Since we are interested in finding a sparse positive definite estimator for the inverse covariance matrix, it is natural to include a barrier function that forces positive definiteness of any minimizer $\boldsymbol{\Theta}$, e.g., $-\log \det(\cdot)$, which has a domain restricted to positive definite matrices. Our optimal estimator $\boldsymbol{\Theta}$ for the inverse covariance matrix $\Sigma^{-1}$ is defined as the following Bregman divergence minimizer

$$\arg\min_{\boldsymbol{\Theta} \in \mathbb{S}_{++}^{p \times p}} \mathrm{B}(\boldsymbol{\Theta}\|\Sigma^{-1}) = \arg\min_{\boldsymbol{\Theta} \in \mathbb{S}_{++}^{p \times p}} -\log \det(\boldsymbol{\Theta}) - \log \det(\Sigma) + \mathrm{tr}(\Sigma(\boldsymbol{\Theta} - \Sigma^{-1})) \tag{1}$$

where $\mathbb{S}_{++}^{p \times p}$ is the set of all symmetric positive definite $p \times p$ matrices. When the covariance matrix $\Sigma$ is unknown the empirical equivalent, the sample covariance $\mathsf{S}$, will be used in its place. The Bregman minimizer (1) does not encourage sparsity of $\boldsymbol{\Theta}$ so a sparsity penalty will be introduced. This penalty also ensures strict convexity.

The empirical Bregman divergence minimizer with barrier function $-\log \det(\cdot)$ and an $\ell_1$ penalty term reduces to

$$\arg\min_{\boldsymbol{\Theta} \in \mathbb{S}_{++}^{p \times p}} \hat{\mathrm{B}}^\lambda(\boldsymbol{\Theta}\|\mathsf{S}^{-1}) = \arg\min_{\boldsymbol{\Theta} \in \mathbb{S}_{++}^{p \times p}} -\log \det(\boldsymbol{\Theta}) + \mathrm{tr}(\boldsymbol{\Theta}\mathsf{S}) + \lambda\|\boldsymbol{\Theta}\|_1 \tag{2}$$

where $\lambda \geq 0$ is a penalty parameter, and $\|\cdot\|_1$ denotes the element-wise $\ell_1$ norm. This can be generalized so that each entry of $\boldsymbol{\Theta}$ can be penalized differently if $\lambda$ is a matrix and using a element-wise product with the norm.

This objective has a unique solution, $\mathbf{\Theta} = (\tilde{\mathsf{S}})^{-1}$, which satisfies

$$\frac{\partial}{\partial \mathbf{\Theta}} \hat{\mathsf{B}}^{\lambda}(\mathbf{\Theta} \| \mathsf{S}^{-1}) = \mathsf{S} - \mathbf{\Theta}^{-1} + \lambda \partial \|\mathbf{\Theta}\|_1 = 0 \qquad (3)$$

where $\partial \|\mathbf{\Theta}\|_1$ is a subdifferential of the $\ell_1$ norm defined in Eq (9) in the Appendix. The solution $(\tilde{\mathsf{S}})^{-1}$ is a sparse positive definite estimator of the inverse covariance matrix $\Sigma^{-1}$, and we can write its inverse explicitly as $\tilde{\mathsf{S}} = \mathsf{S} + \lambda \tilde{\mathsf{Z}}$, where $\tilde{\mathsf{Z}}$ is the unique subdifferential matrix that makes the gradient zero. See [43] or [44] for a more thorough explanation of $\tilde{\mathsf{Z}}$.

Reference [43] shows that for well-conditioned covariances and a certain minimum sample sizes, the estimator $(\tilde{\mathsf{S}})^{-1}$ has many nice properties including having, with high probability, the correct zero and signed non-zero entries and a sum of squared error that converges to 0 as $n, p, s \to \infty$. These properties will allow our method, described in the next section, to attain superior performance over the standard EnKF.

## 3 $\ell_1$ penalized ensemble Kalman filter

Our penalized ensemble Kalman filter, whose pseudo code is shown in Algorithm 1, modifies the EnKF by using a penalized forecast covariance estimator $\tilde{\mathsf{P}}^f$. This penalized estimator is derived from its inverse, which is the minimizer of Eq (2), and can be determined using standard numerical optimization methods [45–47]. Thus, from Section 2.2, it can be explicitly written as $\tilde{\mathsf{P}}^f = \hat{\mathsf{P}}^f + \lambda \tilde{\mathsf{Z}}$, implying that we learn a matrix $\tilde{\mathsf{Z}}$, and use it to modify our sample covariance $\hat{\mathsf{P}}^f$ so that $(\hat{\mathsf{P}}^f + \lambda \tilde{\mathsf{Z}})^{-1}$ is sparse. From this, our modified Kalman gain matrix is

$$\tilde{\mathsf{K}} = (\hat{\mathsf{P}}^f + \lambda \tilde{\mathsf{Z}})\mathsf{H}^T(\mathsf{H}(\hat{\mathsf{P}}^f + \lambda \tilde{\mathsf{Z}})\mathsf{H}^T + \mathsf{R})^{-1} = ((\tilde{\mathsf{P}}^f)^{-1} + \mathsf{H}^T\mathsf{R}^{-1}\mathsf{H})^{-1}\mathsf{H}^T\mathsf{R}^{-1}. \qquad (4)$$

The intuition behind this forecast covariance estimator is that since only a small number of the state variables in the state vector $\mathbf{x}_t$ are conditionally correlated with each other, *the forecast inverse covariance matrix* $(\mathsf{P}^f)^{-1}$ *will be sparse with many zeros in the off-diagonal entries.* Furthermore, since minimizing Eq (2) gives a sparse estimator for $(\mathsf{P}^f)^{-1}$, this sparse estimator will accurately capture the conditional correlations and uncorrelations of the state variables. Thus $\tilde{\mathsf{P}}^f$ will be a much better estimator of the true forecast covariance matrix $\mathsf{P}^f$ because the $\ell_1$ penalty will depress spurious noise in order to make $(\tilde{\mathsf{P}}^f)^{-1}$ sparse, while the inverse of the sample forecast covariance $(\hat{\mathsf{P}}^f)^{-1}$, when it exists, will be non-sparse. In addition, because the estimator of the covariance $\tilde{\mathsf{P}}^f$ is positive definite, standard matrix inversion techniques can be used to convert it to an estimator of the precision matrix.

As in most penalized estimators, $\tilde{\mathsf{P}}^f$ is a biased estimator of the forecast covariance, while the standard covariance is unbiased. But because the forecast distribution is corrected in the analysis step, it is acceptable to take this bias as a trade-off for less variance (sampling errors). A more in-depth study of the consequences of bias in $\ell_1$ penalized inverse covariance matrices and their inverses is described in [44]. Additionally, this bias due to penalization in the inverse covariance can be attributed as naturally occurring variance inflation where the bias on the diagonal of $(\tilde{\mathsf{P}}^f)^{-1}$ is due to the inflation factor $\lambda$. Hence the bias in the forecast covariance estimator is not necessarily disadvantageous. Finally, since we do not assume the state variables interact in the same way over all time, we re-learn the matrix $\tilde{\mathsf{Z}}$ every time the ensemble is evolved forward.

We can choose the penalty parameter $\lambda$ in a systematic fashion by calculating a regularization path, solving Eq (2) for a list of decreasing $\lambda$s, and evaluating each solution with an

information criterion, such as an extended or generalized Akaike information criterion (AIC) or Bayesian information criterion (BIC) [48, 49]. Additionally, if we have knowledge or make additional assumptions about the moments of the ensemble's distribution, we can calculate an optimal penalty parameter λ up to a constant factor (see proof of Theorem 1). Thus, we can refine the penalty parameter by calculating a regularization path for an optimal order penalty function. In Section 4, we describe a practical approach to choosing λ using a free forecast model run like in [37] and using the BIC.

**Algorithm 1** Penalized Ensemble Kalman Filter

**Input:** initial ensemble $A = [a^1, \ldots, a^n]$, measurement operator $H$, process and observation noise covariance matrices $Q$ and $R$, perturbed observation matrix $D_t$ **for** $t \in \{1, \ldots, T\}$ **do**

1) Evolve each ensemble member $a^j$ forward in time $a_0^j = f(a^j) + w^j \quad \forall j \in \{1, \ldots, n\}$ where $w^j \sim N(0, Q)$

2) Calculate the sample covariance for (2) $S = \frac{1}{n-1}(A_0 - \bar{A}_0)(A_0 - \bar{A}_0)^T$ where $\bar{A}_0 = \left(\frac{1}{n}\sum_{j=1}^n a_0^j\right)1^T$

3) Estimate the modified Kalman gain matrix $\tilde{K} = \tilde{P}^f H^T (H\tilde{P}^f H^T + R)^{-1}$ where $\tilde{P}^f = \Theta^{-1}$ and $\Theta$ is the solution to (2)

4) Update the ensemble with the observations $A = A_0 + \tilde{K}(D_t - HA_0)$

5) Predict using the analysis ensemble mean $\tilde{x}_t = \frac{1}{n}\sum_{j=1}^n a^j$

**end for**

**Output:** $\tilde{x}_t, A_t$

## 3.1 Implications on the Kalman gain matrix

The observations $D_t$ come into the EnKF through the ensemble update (step 4 of Algorithm 1), which depends linearly on the Kalman gain matrix $K$. So, having an accurate estimator of the true Kalman gain matrix $K$ will ensure that data is properly merged into the ensemble. Since the true Kalman gain matrix inherits many of the properties of the forecast covariance matrix $P^f$, in view of the relation in step 3 of Algorithm 1, the accuracy of our modified Kalman gain matrix $\tilde{K}$ will depend strongly on the accuracy of the forecast covariance estimator $\tilde{P}^f$.

The quality of the estimated forecast covariance matrix $\tilde{P}^f$ will depend on the structure of the true unknown forecast matrix $P^f$. If $P^f$ is close to singular or contains many entries with magnitudes smaller than the root mean square noise level, it will be difficult to accurately estimate. In the following theorem, we assume that the forecast covariance matrix is well-behaved in the sense that it satisfies standard regularity conditions (incoherence, bounded eigenvalue, sparsity, sign consistency and monotonicity of the tail function) found in [43, 44], which are defined in the Appendix. Additionally, the rate at which the estimator converges depends on the distribution of the ensemble members. For example, if the ensemble members follow a light tailed distribution, more specifically a sub-Gaussian distribution [50], the estimator will typically have a faster rate of convergence than if the ensemble follows a heavy tailed distribution.

**Theorem 1**. *Let $\hat{K}$ be the Kalman gain of the standard EnKF and let $\tilde{K}$ be the modified Kalman gain in the proposed penalized EnKF in Algorithm 1. Under regularity conditions and for the system described in Section 2, when $\lambda \asymp \sqrt{3\log(p)/n}$ for sub-Gaussian ensembles and $\lambda \asymp \sqrt{p^{3/m}/n}$ for ensembles with bounded $4m^{th}$ moments,*

$$\text{Sum of Squared Errors of } \tilde{K} \lesssim \text{Sum of Squared Errors of } \hat{K}$$

*and as long as the sample size is at least $o(n) = 3d^2 \log(p)$ for sub-Gaussian ensembles and $o(n) = d^2 p^{3/m}$ for ensembles with bounded $4m^{th}$ moments,*

$$Sum \; of \; Squared \; Errors \; of \; \tilde{\mathsf{K}} \rightarrow 0 \; with \; high \; probability \; as \; n, p, s \rightarrow \infty.$$

The above theorem gives us a sense of the performance of the modified Kalman gain matrix in comparison to the sample Kalman gain matrix. It shows that with high probability, the modified Kalman gain matrix will have an asymptotically smaller ($\lesssim$) sum of squared error (SSE) than a Kalman gain matrix formed using the sample forecast covariance matrix. Also, for a given state dimensionality $p$, the theorem provides the minimum ensemble size $n$ required for our modified Kalman gain matrix to be a good estimate of the true Kalman gain matrix. The sub-Gaussian criterion, where all moments are bounded, is actually very broad and includes any state vectors with a strictly log-concave density and any finite mixture of sub-Gaussian distributions. However even if not all moments are bounded, the larger the number of bounded fourth-order moments $m$, the smaller the necessary sample size. In comparison, the sample Kalman gain matrix requires $o(n) = p^2$ samples in the sub-Gaussian case, and also significantly more in the other case (see Appendix for exact details). When the minimum sample size for a estimator is not met, good performance cannot be guaranteed because the asymptotic error will diverge to infinity instead of converge to zero. This is why when the number of ensemble members $n$ is smaller than the number of states squared $p^2$, just using the sample forecast covariance matrix is not sufficient. Additionally the theorem also gives us a way to search for the optimal penalty parameter. It tells us how the penalty parameter $\lambda$ should be chosen to scale asymptotically $\asymp$ as a function of the dimensions of the state $p$, the ensemble size $n$, when the ensemble distribution follows either a sub-Gaussian or has $m$ bounded moments.

## 3.2 Implications on the analysis ensemble

It is well known that, due to the additional stochastic noise used to perturb the observations, the covariance of the EnKF's analysis ensemble, $\widehat{\mathrm{Cov}}(\mathsf{A})$ is not equivalent to its analysis covariance calculated by the Gaussian update $\hat{\mathsf{P}}^a = (\mathsf{I} - \hat{\mathsf{K}}\mathsf{H})\hat{\mathsf{P}}^f$. This has led to the development of deterministic variants such as the square root and transform filters, which do have $\widehat{\mathrm{Cov}}(\mathsf{A}) = \hat{\mathsf{P}}^a$. However, in a non-linear system, this update is sub-optimal because it uses a Gaussian approximation of $\mathrm{p}(\mathbf{x}_t|\mathbf{y}_{t-1})$, the actual conditional distribution of forecast ensemble $\mathsf{A}_0$ after $t$ iterations of Algorithm 1. Denote $\mathsf{P}^a$ as the true analysis covariance defined in terms of the posterior state distribution $\mathrm{p}(\mathbf{x}_t|\mathbf{y}_t)$ as

$$\int (\mathbf{x}_t)^2 \mathrm{p}(\mathbf{x}_t|\mathbf{y}_t) d\mathbf{x}_t - \left( \int \mathbf{x}_t \mathrm{p}(\mathbf{x}_t|\mathbf{y}_t) \, d\mathbf{x}_t \right)^2 \tag{5}$$

where $\mathrm{p}(\mathbf{x}_t|\mathbf{y}_t) = \mathrm{p}(\mathbf{y}_t|\mathbf{x}_t)\mathrm{p}(\mathbf{x}_t|\mathbf{y}_{t-1})/\mathrm{p}(\mathbf{y}_t)$ may not be Gaussian. Thus when $\mathrm{p}(\mathbf{x}_t|\mathbf{y}_t)$ is not Gaussian, $\mathrm{E}(\hat{\mathsf{P}}^a) \neq \mathsf{P}^a$ and there will always be an analysis error regardless of whether $\widehat{\mathrm{Cov}}(\mathsf{A}) = \hat{\mathsf{P}}^a$ or not.

In fact, as mentioned in [34], none of the analysis moments of the EnKF are consistent with the true moments, including the analysis mean. This analysis error is present in all methods that do not use unbiased estimates of the posterior distribution, e.g., particle filters. Thus the proposed penalized EnKF will suffer from the same types of biases as other EnKF approximations. However, as established in Theorem 1 and in our experimental results, the effect of these biases on the Kalman gain and on forecasting performance is no worse than other EnKFs.

### 3.3 Computational time and storage issues

The computational complexity of solving for the minimizer of Eq (2) with the GLASSO algorithm from [45] is $O(sp^2)$ because it is a coordinate descent algorithm. Although the final estimator $(\tilde{\mathsf{P}}^f)^{-1}$ is sparse and only requires storing $s + p$ values, the algorithm requires storing $p \times p$ matrices in memory. However, by using iterative quadratic approximations to Eq (2), block coordinate descent, and parallelization, the BIGQUIC algorithm of [46] has computational complexity $O(s(p/k))$ and only requires storing $(p/k) \times (p/k)$ matrices, where $k$ is the number of parallel subproblems or blocks.

The matrix operations for the analysis update $\mathsf{A} = \mathsf{A}_0 + \tilde{\mathsf{K}}(\mathsf{D}_t - \mathsf{HA}_0)$ can also be linear in $p$ if $\mathsf{R}$ is diagonal and $\mathsf{H}$ is sparse (like in banded interpolation matrices) with at most $h \ll r$ non-zero entries in a row. Then $((\tilde{\mathsf{P}}^f)^{-1} + \mathsf{H}^T\mathsf{R}^{-1}\mathsf{H})$ has at most $(s + p + rh^2) \ll p^2$ non-zero entries and can be computed with $O(s + p + rh^2)$ matrix operations. Furthermore $v = \tilde{\mathsf{K}}(\mathsf{D}_t - \mathsf{HA}_0)$ only takes $O(n(s + p + rh^2))$ matrix operations because it is composed of the solutions to the sparse linear systems $((\tilde{\mathsf{P}}^f)^{-1} + \mathsf{H}^T\mathsf{R}^{-1}\mathsf{H})v = \mathsf{H}^T\mathsf{R}^{-1}(\mathsf{D}_t - \mathsf{HA}_0)$, where the right-hand side takes $O(pr^2 + rpn)$ matrix operations to form.

We also point out that EnKF methods do not need to calculate the $p \times p$ sample covariance matrix $\hat{\mathsf{P}}^f$ explicitly, which may be difficult in high-dimensional systems. Instead it only needs to compute on the ensemble matrix, which is only of size $p \times n$, when $n \ll p$. This makes these methods computationally feasible for large $p$; however, they are implicitly using a rank deficient estimator, whose rank is $n$. In order to accurately estimate all dimensions of the system, a full rank estimator is needed of the sample covariance, but this comes with the higher computational cost of order $p^2$ instead of order $pn$. Similarly, EnKF methods also have low memory requirements because they only need to store the $n \times p$ ensemble matrix. In contrast to previous EnKFs, the proposed PEnKF predictor must store the $p \times p$ matrix solution $\Theta$ to (2). However, for sufficiently large sparsity penalty $\lambda$ the solution $\Theta$ and its inverse $\tilde{\mathsf{P}}^f$, will be sparse reducing considerably the memory requirements. The PEnKF can also benefit from dividing the storage up into $k$ blocks of size $(p/k) \times (p/k)$. In any case, memory storage limitations of PEnKF can be rectified by using distributed storage, e.g., cloud virtual disk memory, at the cost of increased overall run time.

## 4 Simulations

In all simulations, we compare the proposed PEnKF to an ensemble Kalman filter where the forecast covariance matrix is localized with a taper matrix generated from equation (4.10) in [51]. The taper matrix parameter for localization $c$ is chosen using the true interactions of the system, so the localization should be close to optimal for simple systems. We use this B-Loc EnKF as the baseline because if the PEnKF can do as well as this filter, it implies that the PEnKF can learn a close to optimal covariance matrix, even without the need to impose known sparsity structure. This would imply that the PEnKF can learn some structure that is not captured by the commonly implemented EnKF method that incorporates localization with a taper matrix.

### 4.0.1 Choosing the penalty parameter

In order to choose the penalty parameter for the PEnKF, we model the state variables in our examples as sub-Gaussian. In this case, we can set $\lambda = c_\lambda \sqrt{R\log(p)/n}$ for some appropriate choice of $c_\lambda$ (see the proof of Theorem 1, where $R$ is the observation noise's variance. To estimate $c_\lambda$, we generate a representative ensemble (which may also be our initial ensemble) using

a free forecast run like in [37] in which a state vector is drawn at random (e.g. from $N(\mathbf{0}, \mathsf{I})$) and evolved forward. The representative ensemble is produced by taking a set of equally spaced points (e.g. every 100th state vector) from the evolution. This ensemble is used to choose $c_\lambda$ from some predefined interval by minimizing the extended Bayesian information criterion (eBIC) of [48] if $p > n$ or the BIC of [52] if $p < n$. If we believe that the penalty parameter should not be constant for all states, e.g. we have multiple types of states in the second simulation scenario, we can search for multiple $c_\lambda$ in a similar fashion.

Of course because Eq (2) has the form of a Gaussian likelihood function, it will only be the correct likelihood if the states are actually Gaussian. As the state transition function $f(x)$ is non-linear, the states will not generally be Gaussian distributed, and hence, the selection of an optimal penalty $\lambda$ using an information criterion like BIC can introduce bias. One way to reduce this bias is to correct our information criterion for the misspecification as in [49]. However, such a correction can be quite difficult. We leave an in-depth exploration of this bias correction problem for future work. Here, we assume the misspecified information criterion is close to the correct information criterion. Our experimental results below show that, despite the bias, it performs well.

### 4.0.2 Metrics

We define the root mean squared error (RMSE) for performance evaluation

$$\mathrm{RMSE}_t = \sqrt{\left(\|\hat{\mathbf{x}}_t - \mathbf{x}_t\|_2\right)^2 / p}, \tag{6}$$

where $\mathrm{RMSE}_t$ is an element of a vector RMSE at any time point $t$, $\mathbf{x}_t$ is a vector of the true hidden state variables, $\hat{\mathbf{x}}_t$ is a filter's predictor for the true state vector, and $\|\cdot\|_2$ is the $\ell_2$ norm. We will refer to statistics such as the mean or median RMSE to be the mean or median of the elements of the RMSE vector.

**4.1 Lorenz 96 system.** The 40-state Lorenz 96 model is one of the most common systems used to evaluate ensemble Kalman filters. The state variables are governed by the following differential equations

$$\frac{dx_t^i}{dt} = \left(x_t^{i+1} - x_t^{i-2}\right)x_t^{i-1} - x_t^i + 8 \qquad \forall i = 1, \cdots, 40 \tag{7}$$

where $x_t^{41} = x_t^1, x_t^0 = x_t^{40}$, and $x_t^{-1} = x_t^{39}$.

We use the following simulation settings. We have observations for the odd state variables, so $\mathbf{y}_t = \mathsf{H}\mathbf{x}_t + \epsilon_t$ where $\mathsf{H}$ is a $20 \times 40$ matrix with ones at entries $\{i, j = 2i - 1\}$ and zeros everywhere else and $\epsilon_t$ is a $20 \times 1$ vector drawn from a $N(\mathbf{0}, 0.5\,\mathsf{I})$. We initialize the true state vector from a $N(\mathbf{0}, \mathsf{I})$ and we assimilate at every $0.4t$ time steps, where $t = 1, \ldots, 2000$. The system is numerically integrated with a 4th order Runge-Kutta method and a step size of 0.01. The main difficulties of this system are the large assimilation time step of 0.4, which makes it significantly non-linear, and the lack of observations for the even state variables.

Since the exact equations of the Lorenz 96 model are fairly simple, it is clear how the state variables interact with each other. This makes it possible to localize with a taper matrix that is almost optimal by using the Lorenz 96 equations to choose a half-length parameter $c$. However, we do not incorporate this information in the PEnKF algorithm, which instead learns interactions by extracting it from the sample covariance matrix. We set the penalty parameter $\lambda = c_\lambda \sqrt{0.5 \log(p)/n}$ by using an offline free forecast run to search for the constant $c_\lambda$ in the range [0.1, 10] as described at the beginning of this section.

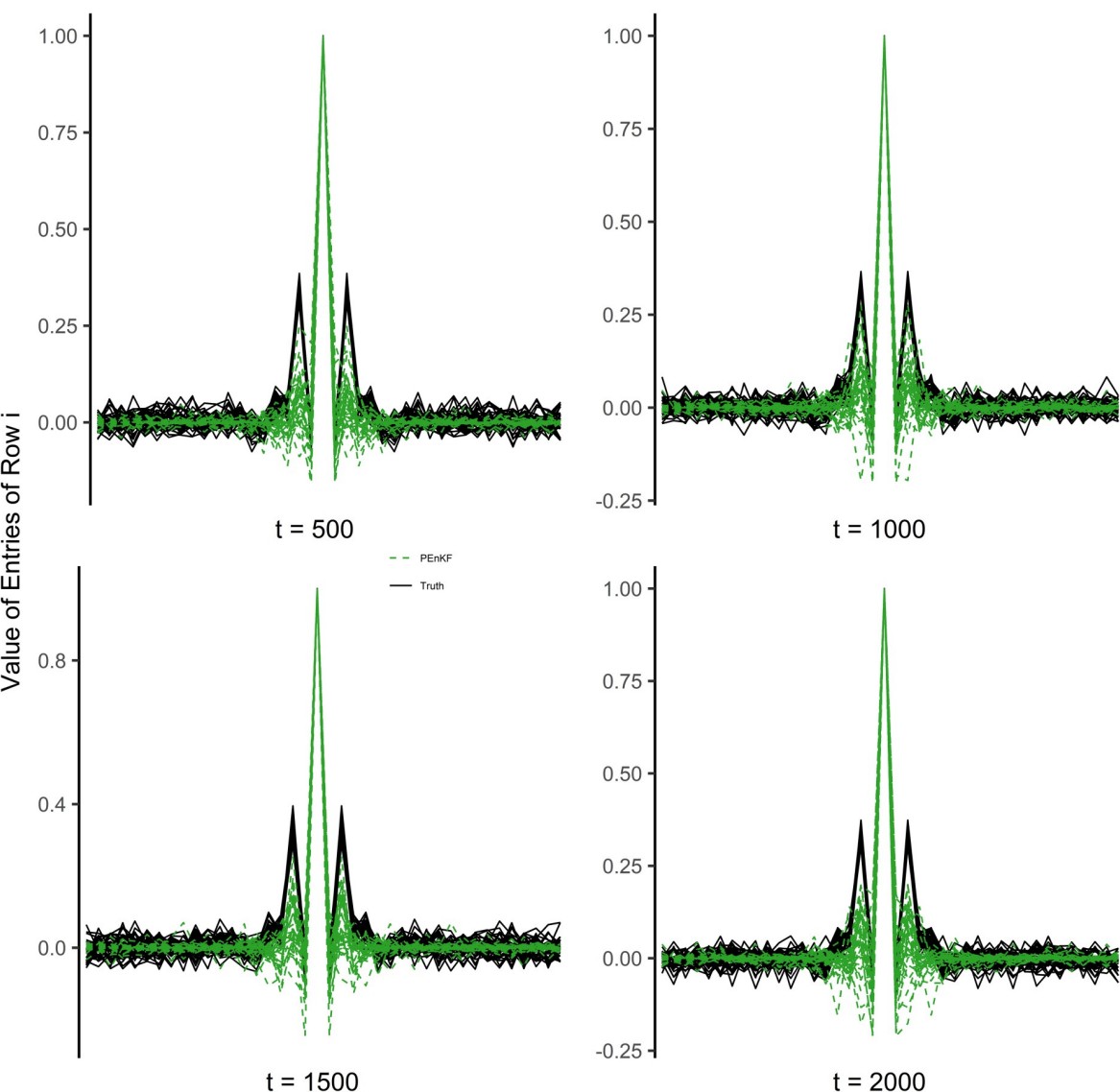

**Fig 1. Each line represents the normalized values of a row of the inverse covariance matrix averaged over 50 trials.** The x-axis shows the entries of a row $i$ ordered from $i − 20$ to $i + 20$, so that the middle of the x-axis is the "diagonal" of the row and the values further from the center of the x-axis are further from the "diagonal". The PEnKF algorithm is successful at identifying that the state variables far away from variable $i$ have no effect on it, even though there are fewer ensemble members than state variables.

We average the PEnKF estimator of the forecast inverse covariance matrix at the time points 500, 1000, 1500, and 2000 for 50 trials with 25 ensembles members, and we compare it to the "true" inverse covariance matrix, which is calculated by moving an ensemble of size 2000 through time. In Fig 1, each line represents the averaged normalized rows of an inverse covariance matrix and the lines are centered at the diagonal. The penalized inverse covariance matrix does a qualitatively good job of capturing the neighborhood information and successfully identifies that any state variables far away from state variable $i$, do not interact with it.

Because the PEnKF is successful at estimating the structure of the inverse covariance matrix and thus the forecast covariance matrix, we expect it will have good performance for estimating the true state variables. We compare the PEnKF to the B-Loc EnKF and other estimators

**Table 1. Mean, median, 10% and 90% quantile of RMSE averaged over 50 trials.** The number in the parentheses is the summary statistics' corresponding standard deviation.

| n = 400 | 10% | 50% | Mean | 90% |
|---|---|---|---|---|
| B-Loc EnKF | 0.580 (.01) | 0.815 (.01) | 0.878 (.02) | 1.240 (.03) |
| PEnKF | 0.538 (.02) | 0.757 (.03) | 0.827 (.03) | 1.180 (.05) |
| n = 100 | 10% | 50% | Mean | 90% |
| B-Loc EnKF | 0.582 (.01) | 0.839 (.02) | 0.937 (.03) | 1.390 (.06) |
| PEnKF | 0.717 (.04) | 0.988 (.04) | 1.067 (.04) | 1.508 (.05) |
| n = 25 | 10% | 50% | Mean | 90% |
| B-Loc EnKF | 0.769 (.04) | 1.668 (.13) | 1.882 (.09) | 3.315 (.11) |
| PEnKF | 0.971 (.03) | 1.361 (.03) | 1.442 (.03) | 2.026 (.04) |
| n = 10 | 10% | 50% | Mean | 90% |
| B-Loc EnKF | 2.659 (.07) | 3.909 (.06) | 3.961 (.05) | 5.312 (.06) |
| PEnKF | 1.147 (.02) | 1.656 (.02) | 1.735 (.02) | 2.437 (.04) |

from [34, 35, 53, 54] by looking at statistics of the RMSE. Note that in order to have comparable statistics to as many other papers as possible, we do not add variance inflation to the B-Loc EnKF (like in [35, 53, 54] and unlike in [34]). Also, like in those papers, we initialize the ensemble from a $N(\mathbf{0}, \mathsf{I})$, and we use this ensemble to start the filters. Note that in this case, the initial ensemble is different than the offline ensemble that we use to estimate the PEnKF's penalty parameter. This is because the initial ensemble is not representative of the system and its sample covariance is an estimator for the identity matrix. The B-Loc EnKF, which is simply called the EnKF in the other papers, is localize by applying a taper matrix where $c = 10$ to the sample covariance matrix.

We show the mean, median, 10%, and 90% quantiles of the RMSE averaged over 50 independent trials for ensembles of size 400, 100, 25, and 10 in Table 1. For 400 ensemble members, the PEnKF does considerably better than the B-Loc EnKF and its relative improvement is larger than that of the XEnKF reported in [53] and similar to those of the NLEAF, EnKPF, and XEnKF reported in [34, 35, 54] respectively. For 100 ensemble members, the PEnKF does do worse than the B-Loc EnKF and EnKPF of [35]; this we suspect may be do to the bias-variance trade-off when estimating the forecast covariance matrix. The PEnKF has the most significant improvement over the B-Loc EnKF in the most realistic regime where we have fewer ensemble members than state variables. For both 25 and 10 ensemble members, the PEnKF does considerably better than the B-Loc EnKF and it does not suffer from filter divergence, which [35] report occurs for the EnKPF at 50 ensemble members.

While it is clear the PEnKF does well even when there are fewer ensemble members than state variables, 40 variables is not enough for the problem to be considered truly high-dimensional. We now consider simulation settings where we increase the dimension of the state space $p$ while holding the number of ensemble members $n$ constant. We initialize the ensemble from the free forecast run and set $\lambda$ and the taper matrix in the same way as in the previous simulations. We examine the mean RMSE averaged over 50 trials and its approximate 95% confidence intervals in the Fig 2. The mean RMSE of the PEnKF is significantly smaller than the mean RMSE of the B-Loc EnKF for all $p$. Additionally the confidence intervals of the mean RMSE are much narrower than the ones for the B-Loc EnKF. This suggest that there is little variability in the PEnKF's performance, while the B-Loc EnKF's performance is more dependent on the trial, with some trials being "easier" for the B-Loc EnKF than others.

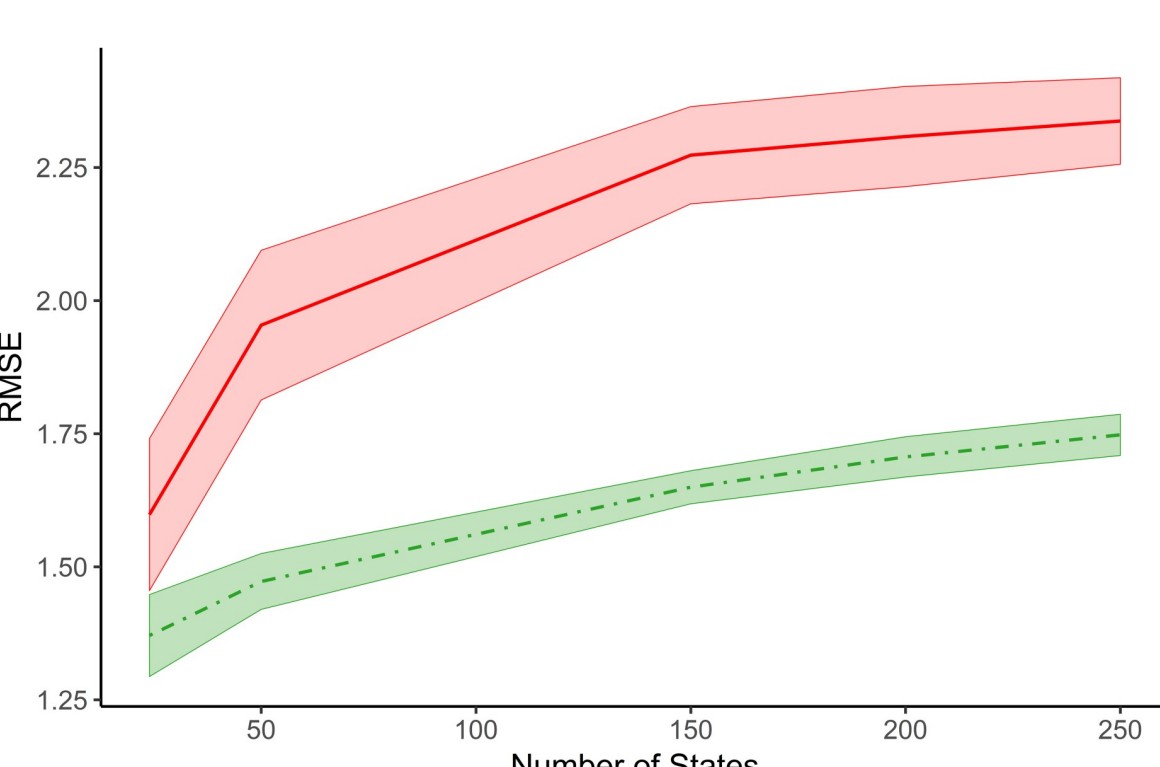

**Fig 2. The RMSE of the B-Loc EnKF and PEnKF over 50 trials using 25 ensemble members.** The darker lines of each linetype are the mean and the colored areas are the 95% confidence intervals. There is clear separation between the RMSE of the two filters with the PEnKF's error being significantly smaller.

### 4.2 Modified shallow water equations system

While the Lorenz 96 system shows that the PEnKF has strong performance because it is successful at reducing the sampling errors and is capable of learning the interactions between state variables, the system is not very realistic in that all state variables are identical and the relationships between state variables are very simplistic. We now consider a system based on the modified shallow water equations of [55], which models cloud convection with fluid dynamics equations, but is substantially less computationally expensive than full scale numerical weather prediction models. The system has three types of state variables: fluid height, rain content, and horizontal wind speed, which are shown in (S1)(S2)(S3) of the Appendix.

To generate observations from this system we use the R package "modifiedSWEQ" created by the authors of [56], and use the same simulation settings as in [37]. So we always observe the rain content, but wind speed is only observed at locations where it is raining and fluid height is never observed. Explicitly for the R function *generate.xy*(), we use $h_c = 90.02$, $h_r = 90.4$ for the cloud and rainwater thresholds, a 0.005 rain threshold, $\sigma_r = 0.1$, $\sigma_u = 0.0025$ to be the standard deviation of the observation noise for rain and wind respectively, and $\mathsf{R} = \mathrm{diag}([R_r^2 = 0.025^2 \ R_u^2 = \sigma_u^2])$ to be the estimated diagonal noise covariance matrix. All other parameters are just the default ones in the function. The initial ensemble is drawn from a free forecast run with 10000/60 time-steps between each ensemble member. We give a snapshot of the system at a random time point in Fig 3. There are $p = 300$ state variables for each type, making the state space have 900 dimensions and we assimilate the system every 5 seconds

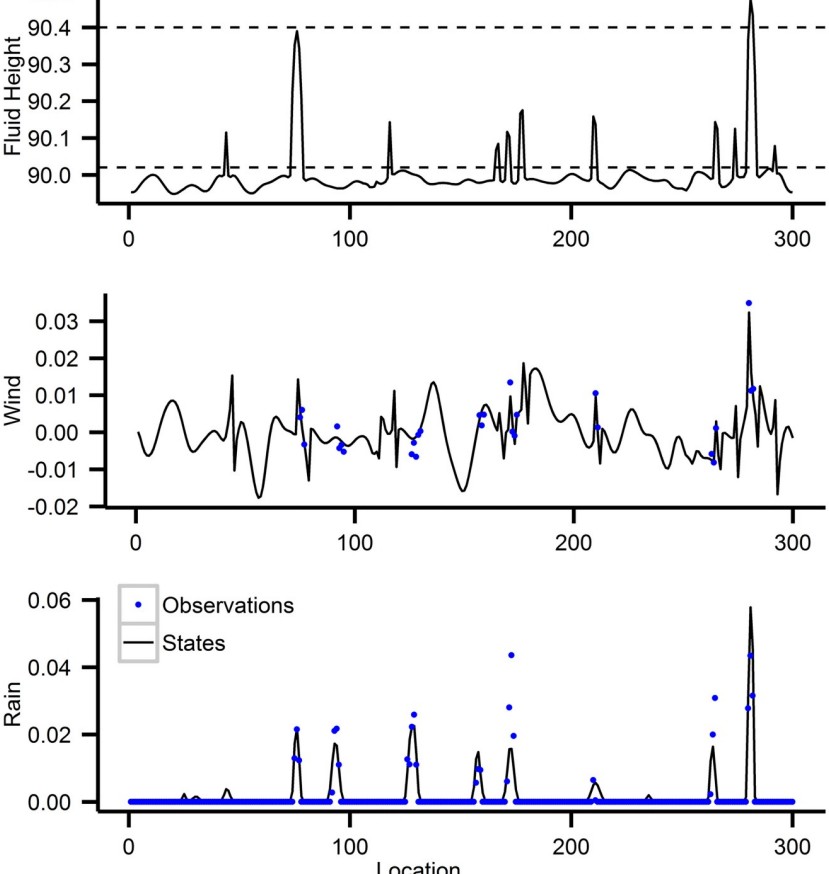

**Fig 3. Fluid height, rain, and wind at 300 different locations at an instance of time.** The blue dots are observations; rain is always observed, wind is only observed when the rain is non-zero, fluid height is never observed. The dashed lines in fluid height are the cloud and rainwater thresholds.

for a total time period of 6 hours. Like in [37], we choose to use only 50 ensemble members and we do not perturb rain observations that are 0, because at these points there is no measurement noise.

The B-Loc EnKF uses a $3p \times 3p$ taper matrix with $c = 5$, however the entries off the $p \times p$ block diagonals are depressed (they are multiplied by 0.9). The NAIVE-LEnKPF uses the same settings as in [37], so a localization parameter of 5km, which gives the same taper matrix as the one used in the B-Loc EnKF, and an adaptive $\gamma$ parameter. For the PEnKF, we set the penalty parameter to be a $3p \times 3p$ matrix, $\Lambda = c_\lambda \sqrt{\lambda_R \lambda_R^T \log(3p)/n}$, where the first $p$ entries of the vector $\lambda_R$ are reference units and the rest are to scale for the perturbation noise of the different state types. So the first $p$ are 1 (reference) for fluid height, the second $p$ are $R_u$ for wind, and the last $p$ are $R_r$ for rain. We choose the constant $c_\lambda$ with eBIC like before and search in the range [.005, 1].

Fig 4 shows the mean and approximate 95% confidence intervals of the RMSE for fluid height, wind speed, and rain content over 6 hours of time using 50 trials. The mean RMSE for all three filters are well within each others' confidence intervals for the fluid height and wind variables. For the rain variables, the mean RMSE of neither the B-Loc EnKF nor the NAIVE-LEnKPF are in the PEnKF's confidence intervals and the mean RMSE of the PEnKF is on the

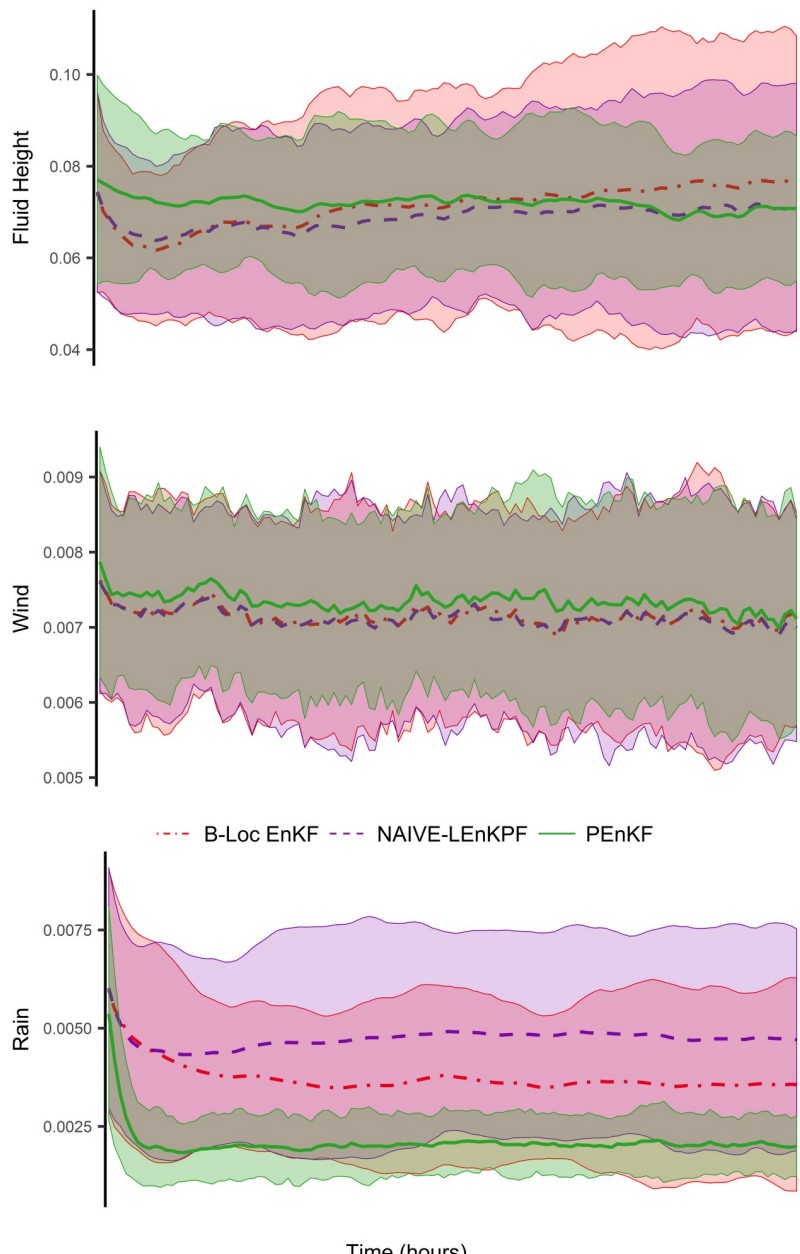

**Fig 4. The RMSE of the B-Loc EnKF, NAIVE-LEnKPF, and PEnKF over 50 trials.** The darker lines of each linetype are the mean and the colored areas are the 95% confidence intervals. All three filters are pretty indistinguishable except for the PEnKF's rain error, which is statistically smaller than the others.

boundary of the other two models' confidence intervals. This strongly suggests that the PEnKF's rain error is statistically smaller than the rain errors of the other two filters. Since this simulation is not as simple as the previous ones, the interactions between the state variables are most likely not as effectively captured by the taper matrix or other localization methods, and the results from this simulation suggest that the PEnKF is learning more accurate interactions for the rain variables. We do not show the results of the BLOCK-LEnKPF of [37] because

the algorithm suffered from filter divergence in 27 of the 50 trials, and in the trials where it did not fail, it performed very similar to the NAIVE-LEnKPF.

## 5 Discussion

We propose a new algorithm based on an unstructured ensemble Kalman filter that is designed for superior performance in non-linear high dimensional systems when dependency structure in the state vector is unknown. This algorithm we call the penalized ensemble Kalman filter because it uses the popular statistical concept of penalization/regularization in order to make the problem of estimating the forecast covariance matrix well-defined (strictly convex). This in turn both decreases the sampling errors (variance) in the forecast covariance estimator by trading it off for bias and prevents filter divergence by ensuring that the estimator is positive definite. The PEnKF is computationally efficient in that it is not significantly slower than the standard EnKF algorithms and easy to implement since it only adds one additional step. This step uses the well-established GLASSO algorithm, available in almost any scientific computing language, for sparse estimation of inverse covariance matrices. We give theoretical results that prove that the Kalman gain matrix constructed from this estimator will converge to the population Kalman gain matrix under the non-simplistic asymptotic case of high-dimensional scaling, where the sample size and the dimensionality increase to infinity.

Through simulations, we show that the PEnKF can do at least as well as, and sometimes better than, localized filters that use much more prior information. We emphasize that by doing just as well as the B-Loc EnKF, which has a close to optimal taper matrix, the PEnKF is effectively correctly learning the structure of interactions between the state variables. Thus the PEnKF is able to infer from the ensemble the true conditional correlation between the states as opposed to having it or the correlation pre-defined. In a non-simulation setting where there is possibly more uncertainty in the prior knowledge of the interactions between state variables, correct localization is much more difficult, making any localized filter's performance likely sub-optimal. In contrast, since the PEnKF does not use any explicit prior knowledge, its performance will not differ in this way between simulations and real-life situations. The more complicated simulation, based on the modified shallow water equations, highlights this advantage of the PEnKF through its substantial superior performance in estimating the hidden states of the rain variables. This is because the relationship between states from different types (e.g. rain and fluid height) is far less obvious than the relationship between states of the same type, which is heavily influenced by physical location. Another feature of the approach is that it seems to require less variance inflation. None was applied to any algorithm in our comparison, but the PEnKF approach never collapsed. The penalization of the inverse covariance actually produces a slight inflation on the diagonal of the covariance, which seems to help in this regard.

We propose a simple, but effective BIC penalty method to search for a penalty parameter for the PEnKF that performs well in the simulations. However, it is well known that such simple methods are biased and other more sophisticated methods can result in penalties that are closer to optimal. Since the PEnKF can be sensitive to the penalty parameter, we think that it will be worthwhile to investigate improved penalty parameter selection methods for the PEnKF. We also think that such an improvement may be very challenging due to the more complex mapping imposed by the $\ell_1$ penalty that we use to enforce sparsity. This topic is a very active area of research in theoretical statistics, which may enable advances enabling improved PEnKF methods. Additionally, there have been interesting new developments in the mathematical behavior of the modes of a dynamical system; specifically in [57] where they show that the number of ensemble members $n$ needs only to be of the size of the growing and

neutral modes in the dynamical system because the decaying modes do not hamper forecasting. This role of the modes of a dynamical system in our more general non-linear dynamical system model would be a promising avenue for future work.

# 6 Appendix

**Definition**.

*(D1) $\sigma_{\min}(\mathsf{A})$ and $\sigma_{\max}(\mathsf{A})$ denote the minimum and maximum singular values of any matrix $\mathsf{A}$.*

*(D2) The spectral $\|\cdot\|_2$ and Frobenius $\|\cdot\|_F$ norms are submultiplicative $\|\mathsf{AB}\| \leq \|\mathsf{A}\|\|\mathsf{B}\|$ and unitary invariant $\|\mathsf{AU}\| = \|\mathsf{U}^T\mathsf{A}\| = \|\mathsf{A}^T\|$ where $\mathsf{UU}^T = \mathsf{I}$. So*
$$\|\mathsf{AB}\|_F = \|\mathsf{AUDV}^T\|_F = \|\mathsf{AUD}\|_F \leq \|\mathsf{AU}\sigma_{\max}(\mathsf{D})\|_F = \|\mathsf{AU}\|_F\|\mathsf{D}\|_2 = \|\mathsf{A}\|_F\|\mathsf{B}\|_2$$

*(D3) $\|\mathsf{A}^{-1}\|_2 = \sigma_{\max}(\mathsf{A}^{-1}) = 1/\sigma_{\min}(\mathsf{A})$*

*(D4) $\mathsf{K}$ and $\tilde{\mathsf{K}}$ can be decomposed like*

$$
\begin{aligned}
\mathsf{K} \quad &= \mathsf{P}^f\mathsf{H}^T(\mathsf{HP}^f\mathsf{H}^T + \mathsf{R})^{-1} \\
&= \mathsf{P}^f\mathsf{H}^T(\mathsf{R}^{-1} - \mathsf{R}^{-1}\mathsf{H}((\mathsf{P}^f)^{-1} + \mathsf{H}^T\mathsf{R}^{-1}\mathsf{H})^{-1}\mathsf{H}^T\mathsf{R}^{-1}) \\
&= (\mathsf{P}^f - \mathsf{P}^f((\mathsf{P}^f)^{-1}(\mathsf{H}^T\mathsf{R}^{-1}\mathsf{H})^{-1} + \mathsf{I})^{-1})\mathsf{H}^T\mathsf{R}^{-1} \\
&= (\mathsf{P}^f - \mathsf{P}^f((\mathsf{H}^T\mathsf{R}^{-1}\mathsf{H})^{-1} + \mathsf{P}^f)^{-1}\mathsf{P}^f)\mathsf{H}^T\mathsf{R}^{-1} \\
&= ((\mathsf{P}^f)^{-1} + \mathsf{H}^T\mathsf{R}^{-1}\mathsf{H})^{-1}\mathsf{H}^T\mathsf{R}^{-1}.
\end{aligned}
\tag{8}
$$

*(D5) $\mathcal{E}$ is the edge set corresponding to non-zeros in $(\mathsf{P}^f)^{-1}$ and $\mathcal{E}^c$ is its complement. $\Gamma$ is the Hessian of* Eq (2).

*(D6) $\partial\|\Theta\|_1$ can be any number between -1 and 1 where $\Theta$ is 0 because the derivative of an absolute value is undefined at zero. Thus, it is the set of all matrices $\mathsf{Z} \in \mathbb{S}^{p \times p}$ such that*

$$
Z_{ij} = \left\{
\begin{array}{ll}
\operatorname{sign}(\Theta_{ij}) & \text{if} \quad \Theta_{ij} \neq 0 \\
\in [-1, 1] & \text{if} \quad \Theta_{ij} = 0\,.
\end{array}
\right.
\tag{9}
$$

*(D7) A bounded $4m^{th}$ moment is the highest fourth-order moment of a random variable that is finite, where m is the number of fourth-order moments.*

**Lemma 1.1**. *Because $\mathsf{H}$ is a constant matrix, it does not affect the asymptotic magnitude of the modified or sample Kalman gain matricies under any norm.*

*Proof of Lemma 1.1.* $\|\tilde{\mathsf{K}}\| \asymp \|\mathsf{H}\|$ $\|\tilde{\mathsf{K}}\| \asymp \|\mathsf{H}\tilde{\mathsf{K}}\|$ *under any norm where* $\mathsf{H}\tilde{\mathsf{K}}$

$$
\begin{aligned}
&= \mathsf{H}\tilde{\mathsf{P}}^f\mathsf{H}^T(\mathsf{H}\tilde{\mathsf{P}}^f\mathsf{H}^T + \mathsf{R})^{-1} = (\mathsf{I} + \mathsf{R}(\mathsf{H}\tilde{\mathsf{P}}^f\mathsf{H}^T)^{-1})^{-1} \\
&= \mathsf{RR}^{-1}(\mathsf{R}^{-1} + (\mathsf{H}\tilde{\mathsf{P}}^f\mathsf{H}^T)^{-1})^{-1}\mathsf{R}^{-1} \\
&= \mathsf{I} - \mathsf{R}(\mathsf{H}\tilde{\mathsf{P}}^f\mathsf{H}^T + \mathsf{R})^{-1}
\end{aligned}
\tag{10}
$$

The same argument holds for $\hat{\mathsf{K}}$, where $(\mathsf{H}\hat{\mathsf{P}}^f\mathsf{H}^T)^{-1}$ is the pseudoinverse if the inverse does not exist.

**Assumptions**. *The following assumptions are necessary for the minimizer of* Eq (2) *to have good theoretical properties,* [43]. *Thus we assume they are true for the theorem.*

*(A1) There exists some $\alpha \in (0, 1]$ such that $\max\limits_{e \in \mathcal{E}^c} \|\Gamma_{e\mathcal{E}}(\Gamma_{\mathcal{E}\mathcal{E}})^{-1}\|_1 \leq (1 - \alpha)$.*

*(A2) The ratio between the maximum and minimum eigenvalues of $\mathsf{P}^f$ is bounded.*

*(A3) The maximum $\ell_1$ norms of the rows of $\mathsf{P}^f$ and $(\Gamma_{\mathcal{E}\mathcal{E}})^{-1}$ are bounded.*

*(A4) The minimum non-zero value of $(\mathsf{P}^f)^{-1}$ is $\Omega(\sqrt{\log(p)/n})$ for a sub-Gaussian state vector and $\Omega(\sqrt{p^{3/m}/n})$ for state vectors with bounded $4m^{th}$ moments.*

*Our assumptions are stronger than necessary, and it is common to allow the error rates to depend on the bounding constants above, but for simplicity we give the error rates only as a function of the dimensionality $n$, $p$ and sparsity $s$, $d$ parameters.*

*Proof of Theorem 1.* From [58] and [43], we know that for sub-Gaussian random variables and those with bounded $4m^{th}$ moments respectively, the SSE of the sample covariance matrix are

$$\begin{cases} O(p^2/n) \\ \\ O((\log_2 \log_2(p))^4 p(p/n)^{1-1/m}) \end{cases} \tag{11}$$

and with high probability and the SSE of $(\tilde{\mathsf{P}}^f)^{-1}$ are

$$\begin{cases} O(3(s+p)\log(p)/n) & \text{for} \quad \lambda \asymp \sqrt{3\log(p)/n} \\ \\ O((s+p)p^{3/m}/n) & \text{for} \quad \lambda \asymp \sqrt{p^{3/m}/n} \end{cases} \tag{12}$$

with probability $1 - 1/p$.

$$\begin{aligned}
\|\mathsf{H}\hat{\mathsf{K}} - \mathsf{H}\mathsf{K}\|_F^2 &= \|\mathsf{R}(\mathsf{H}\mathsf{P}^f\mathsf{H}^T + \mathsf{R})^{-1} - \mathsf{R}(\mathsf{H}\hat{\mathsf{P}}^f\mathsf{H}^T + \mathsf{R})^{-1}\|_F^2 \\
&= \|\mathsf{R}((\mathsf{H}\hat{\mathsf{P}}^f\mathsf{H}^T + \mathsf{R})^{-1}((\mathsf{H}\hat{\mathsf{P}}^f\mathsf{H}^T + \mathsf{R}) - (\mathsf{H}\mathsf{P}^f\mathsf{H}^T + \mathsf{R}))(\mathsf{H}\mathsf{P}^f\mathsf{H}^T + \mathsf{R})^{-1})\|_F^2 \\
&= \|\mathsf{R}(\mathsf{H}\hat{\mathsf{P}}^f\mathsf{H}^T + \mathsf{R})^{-1}\mathsf{H}(\hat{\mathsf{P}}^f - \mathsf{P}^f)\mathsf{H}^T(\mathsf{H}\mathsf{P}^f\mathsf{H}^T + \mathsf{R})^{-1}\|_F^2 \\
&\leq \|\mathsf{R}(\mathsf{H}\hat{\mathsf{P}}^f\mathsf{H}^T + \mathsf{R})^{-1}\mathsf{H}\|_2^2 \|(\hat{\mathsf{P}}^f - \mathsf{P}^f)\|_F^2 \|\mathsf{H}^T(\mathsf{H}\mathsf{P}^f\mathsf{H}^T + \mathsf{R})^{-1}\|_F^2
\end{aligned} \tag{13}$$

So, the second factor has the rates in Eq (11) and the final factor is a constant. The first factor is also a constant because

$$\begin{aligned}
\|\mathsf{R}(\mathsf{H}\hat{\mathsf{P}}^f\mathsf{H}^T + \mathsf{R})^{-1}\mathsf{H}\|_2^2 &\leq \|(\mathsf{H}\hat{\mathsf{P}}^f\mathsf{H}^T\mathsf{R}^{-1} + \mathsf{I})^{-1}\|_2^2 \|\mathsf{H}\|_2^2 \\
&= \|\mathsf{H}\|_2^2 / (\sigma_{\min}(\mathsf{H}\hat{\mathsf{P}}^f\mathsf{H}^T\mathsf{R}^{-1} + \mathsf{I}))^2 \leq \|\mathsf{H}\|_2^2.
\end{aligned} \tag{14}$$

Thus $\|\mathsf{H}\hat{\mathsf{K}} - \mathsf{H}\mathsf{K}\|_F^2$ also has the rates in Eq (11) and from Lemma 1.1, $\|\hat{\mathsf{K}} - \mathsf{K}\|_F^2$ does too.

$$\begin{aligned}
&\|\tilde{\mathsf{K}} - \mathsf{K}\|_F^2 \\
&= \|(((\tilde{\mathsf{P}}^f)^{-1} + \mathsf{H}^T\mathsf{R}^{-1}\mathsf{H})^{-1} - ((\mathsf{P}^f)^{-1} + \mathsf{H}^T\mathsf{R}^{-1}\mathsf{H})^{-1})\mathsf{H}^T\mathsf{R}^{-1}\|_F^2 \\
&= \|(((\mathsf{P}^f)^{-1} + \mathsf{H}^T\mathsf{R}^{-1}\mathsf{H})^{-1}(((\mathsf{P}^f)^{-1} + \mathsf{H}^T\mathsf{R}^{-1}\mathsf{H}) \\
&\quad - ((\tilde{\mathsf{P}}^f)^{-1} + \mathsf{H}^T\mathsf{R}^{-1}\mathsf{H}))((\tilde{\mathsf{P}}^f)^{-1} + \mathsf{H}^T\mathsf{R}^{-1}\mathsf{H})^{-1})\mathsf{H}^T\mathsf{R}^{-1}\|_F^2 \\
&\leq \|((\mathsf{P}^f)^{-1} + \mathsf{H}^T\mathsf{R}^{-1}\mathsf{H})^{-1}\|_F^2 \|(\mathsf{P}^f)^{-1} - (\tilde{\mathsf{P}}^f)^{-1}\|_F^2 \|\tilde{\mathsf{K}}\|_2^2
\end{aligned} \tag{15}$$

The first term is a constant and the second term has the rates in Eq (12). The final term is also a constant because $\|\tilde{\mathsf{K}}\|_2^2 \asymp \|\mathsf{H}\tilde{\mathsf{K}}\|_2^2 = 1/\sigma_{\min}(\mathsf{I} + \mathsf{R}(\mathsf{H}\tilde{\mathsf{P}}^f\mathsf{H}^T)^{-1}) \leq 1$. Thus $\|\tilde{\mathsf{K}} - \mathsf{K}\|_F^2$ also has the rates in Eq (12).

**Shallow water equations**. The following equations are the modified shallow water equations described in [55].

(S1) $\frac{\partial u}{\partial t} + u\frac{\partial u}{\partial x} + \frac{\partial(\phi + gH_0 r)}{\partial x} = K\frac{\partial^2 u}{\partial x^2} + \bar{u}\frac{\partial}{\partial x}\left(\exp\{-(x-x_n)^2/l^2\}\right)$

(S2) $\frac{\partial h}{\partial t} + \frac{\partial(uh)}{\partial x} = K\frac{\partial^2 h}{\partial x^2}$

(S3) $\frac{\partial r}{\partial t} + u\frac{\partial r}{\partial x} = K_r\frac{\partial^2 r}{\partial x^2} - \alpha r - \begin{cases} \beta\frac{\partial u}{\partial x}, & Z > H_r \text{ and } \frac{\partial u}{\partial x} < 0 \\ 0, & otherwise \end{cases}$

which are centered around location $x_n$ with length scale l, and where u is the fluid velocity and $\bar{u}$ is its amplitude, h is the fluid depth, and r is the mass fraction of rain water. The geopotential φ is based on the height of the fluid surface and defined as

$\phi = \begin{cases} \phi_c + gH, & Z > H_r \text{ and } \dfrac{\partial u}{\partial x} < 0 \\ g(H + h), & otherwise \end{cases}$ where H is the height of the topography with

thresholds $H_c$ and $H_r$ for convection and rainwater respectively, and $H_0$ is the initial absolute fluid layer height. α and β are physical parameters that can be varied to produce realistic space and time scales for the cloud models.

## Author Contributions

**Conceptualization:** Elizabeth Hou, Earl Lawrence, Alfred O. Hero.

**Formal analysis:** Elizabeth Hou.

**Funding acquisition:** Earl Lawrence, Alfred O. Hero.

**Methodology:** Elizabeth Hou, Earl Lawrence, Alfred O. Hero.

**Writing – original draft:** Elizabeth Hou, Earl Lawrence.

**Writing – review & editing:** Elizabeth Hou, Earl Lawrence, Alfred O. Hero.

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
