## [Decision Letter · Decision Letter 0]

23 Jul 2020

PONE-D-20-14752

Penalized Ensemble Kalman Filters for High Dimensional Non-linear Systems

PLOS ONE

Dear Dr. Hou,

Thank you for submitting your manuscript to PLOS ONE. After careful consideration, we feel that it has merit but does not fully meet PLOS ONE’s publication criteria as it currently stands. Therefore, we invite you to submit a revised version of the manuscript that addresses the points raised during the review process.

You will see that the reviewers offer several constructive suggestions to improve the readability of the manuscript, in particular to disclose the potential of the proposed method and its reproducibility. I am confident you will be able to account for all these comments. The reviewers and myself also found the abstract needs rewording, to provide specific description of the PEnKF along with its strength and weaknesses. I also recommend you not to use the expression "high-dimensional problems" for the two applications examples whose size is not comparable to operational geophysical data assimilation problems. I suggest you to add a discussion on the feasibility of PEnKF implementation for real problems (e.g. numerical weather prediction or operational oceanography or related dicipline) in the last section. The quality of figures requires improvement and I also found some references that cannot be found online: please also check these aspects in the revised version of the manuscript.

We look forward to receiving your revised manuscript.

Kind regards,

Andrea Storto

Academic Editor

PLOS ONE

Journal Requirements:

2.In your Data Availability statement, you have not specified where the minimal data set underlying the results described in your manuscript can be found. PLOS defines a study's minimal data set as the underlying data used to reach the conclusions drawn in the manuscript and any additional data required to replicate the reported study findings in their entirety. All PLOS journals require that the minimal data set be made fully available. For more information about our data policy, please see http://journals.plos.org/plosone/s/data-availability.

3.Thank you for stating the following in your Competing Interests section: 

[No].

Reviewers' comments:

Reviewer's Responses to Questions

**Comments to the Author**

1. Is the manuscript technically sound, and do the data support the conclusions?

Reviewer #1: Partly

Reviewer #2: No

Reviewer #3: Yes

2. Has the statistical analysis been performed appropriately and rigorously? 

Reviewer #1: Yes

Reviewer #2: No

Reviewer #3: Yes

3. Have the authors made all data underlying the findings in their manuscript fully available?

Reviewer #1: Yes

Reviewer #2: Yes

Reviewer #3: Yes

4. Is the manuscript presented in an intelligible fashion and written in standard English?

Reviewer #1: Yes

Reviewer #2: No

Reviewer #3: Yes

5. Review Comments to the Author

Reviewer #1: This is a very interesting paper on reducing noisy covariance estimates in Ensemble Kalman filters. The paper is well-written in general, but it is perhaps too easy to lose non-mathematicians. Assuming this new scheme is meant to be used in practice the authors should provide an easier access for practitioners. Furthermore, the claims on its applicability for real-world systems are perhaps overstated. I suggest a modification of the paper along the following lines.

1. Explain in the abstract how the new filter is different from localization. At the moment the abstract contains nothing new.

2. On particle filters, I suggest the authors have a look at the latest on particle filtering in high-dimensional systems, such as Van Leeuwen, P.J., L. Nerger, R. Potthast, S. Reich, and H.R. Kunsch (2019) Particle Filters for Geoscience applications: A Review, Q.J.Royal Meteorol. Soc., doi: 10.1002/qj.3551. No need to reference it, but there is more than equal weights and mixtures with EnKF in the particle filter literature.

3. Lines 70-95: The fact that a conditionally sparse correlation matrix has a sparse inverse is extremely important and a fact I wasn't aware of, as I'm sure many in the data-assimilation community are not. Please provide proof or reference.

4. Algorithm 1 still uses perturbed observations, while it has been shown that perturbing H(x) is the correct thing to do, and the better thing to do when the system is nonlinear, see e.g. Van Leeuwen, P.J., (2020) A consistent interpretation of the stochastic version of the Ensemble Kalman Filter, Q.J. Royal Meteorol. Soc., doi: 10.1002/qj.3819

5. Alg 1: Perhaps mention here that an EnKF scheme will not need to calculate P^f explicitly in high-dimensional systems but instead works with the ensemble matrix of size n*p.

6. Please number all equations. In the equation below eq 1, please define B^{\\lambda}, it seems that it is a Bergman divergence and the equation below eq (1) is actually its definition? Furthermore, the appendix is missing, or is the 'Definition' the start of the appendix? . I suggest to expand section 2.2, it is most important!

7. Define the deformed = sign in Theorem 1. Note PLOS ONE is not a maths or stats journal, but tries to be broader than that. What is a sub-Gaussian ensemble?

8. line 175: p is the dimension of the state, not the number of states.

9. lines 183-187: The authors seem to forget that the number n needed depends strongly on the system dynamics: in simple terms n should be of the size of the growing and neutral modes in the dynamical system, simply because the decaying modes do not hamper a forecast. See e.g. SIAM/ASA J. Uncertainty Quantification, 5(1), 304–333.

Degenerate Kalman Filter Error Covariances and Their Convergence onto the Unstable Subspace Marc Bocquet, Karthik S. Gurumoorthy, Amit Apte, Alberto Carrassi, Colin Grudzien, and Christopher K. R. T. Jones https://doi.org/10.1137/16M1068712

10. Line 188: Pr(..) is typically used to denote probability, while the authors mean probability density here. Perhaps change notation.

11. Section 3.3: what is q? Is that r?

12. Section 3.3: On the storage, assume the computer can store 100*dim(x) variables (that is reality for numerical weather prediction). And suppose we have 10,000 blocks. Than a matrix of size (p/k)*(p/k) = 10,000 dim(x)^2 /10^8 = dim(x)^2/10^4. For a typical system dim(x) = 10^9 or 10^10, so there is no way one can store this. In fact, even s can be too large to store. That is why all ensemble methods work with the ensemble matrix of size n*p. Please add something like this to the discussion of the computational burden.

13. Perhaps provide pseudo code for the new filter. That would help sell it strongly...!

14. Please describe what is shown in fig 1 more accurately. The caption is confusing and incomplete, and the quality of the figure is not ok. Furthermore, spurious correlations are still 5%, which would ruin a weather forecast. Please comment.

15. Fig 2: How large is n? (This is a remarkable result, great!)

16. Section 4.2: Please provide model equations, e.g. in an appendix.

17. Section 5: Can the authors explain the use of the word 'learning' when discussing the PEnKF? I seem to be missing how information from one data assimilation step is propagating to the other, other than via the ensemble itself. All methods have that in common, so that cannot be the learning part.

18. Line 375: There are many ways to infer the performance of a data assimilation system applied to real systems. And there is always a lot of scientific knowledge about the system under study. Because of this, I find the 'no ground truth' argument less convincing. More convincing is that despite this knowledge a lot of tuning is needed to make things work. The PEnKF might reduce the amount of tuning needed. And please don't use the wording 'oracle information'; as mentioned we always have prior information so we should always use it.

19. Following on from the previous point, the PEnKF also needs tuning of the lambda matrix for the L1 optimization. How is that different from tuning a localization lengthscale?

20. Line 385: One has to be careful, the authors mention that the PEnKF has a slight implicit inflation but in their comparison they do not allow other methods, that do not have this implicit inflation, to have some explicit inflation. It is practical knowledge that a slight inflation can often help any EnKF tremendously. So one might consider the comparison somewhat unfair. Also, note that most operational weather prediction run with some kind of adaptive inflation method based on the performance of the data assimilation.

21. Proof of Theorem 1: Replace 'term' by 'factor'.

Reviewer #2: Please, find attached my comments. ................................................................................................. .................................................................................................

Reviewer #3: This manuscript proposes a method which allows the application of ensemble Kalman filters in high-dimensional systems keeping accuracy and stability, and at the same time allowing for realistic ensemble sizes much smaller than the actual size of the system. To do this it uses a penalisation based on ideas from lasso. The idea is find an estimator of the inverse of the covariance matrix subject to some constrains like sparcity. This is

I think this is a very interesting contribution to the KF and DA literature. It is very well written. It presents a good literature review, nice theoretical derivations and some numerical experiments. I think this will be acceptable for publication after some minor reviews.

-- Minor Comments:

- The sub-Gaussian criterion described in line 177 seems quite important for theorem 1. In my first read it was somehow lost, and then I had to go back and read more carefully. Can you emphasise it somehow? Or maybe even state this criterion before the theorem.

- I would really avoid using the term TAPER-EnKF, since I have never seen it before in the EnKF literature. What you are describing is an EnKF with B-localisation, or localisation in model space.

-- Comments:

- In line 234 you mention equation (1) is not a likelihood unless the states are Gaussian. Is it really as restrictive as this? Or can the states belong to certain class of distributions?

- Your approach is quite interesting since you focus on the construction of inv(Pf) rather than Pf itself, and you ask from some requirements from inv(Pf).

Let's say you have a real Pf and two estimators: one for Pf and the other for its inverse. How does the sparsity in one relate to the other? I am just reading line 130 where you mention the 'intuition' about inv(Pf) being sparse and diagonal dominant. It is usually this assumption but on Pf what is used. How do both relate?

- In many applications localisation is done differently in different physical dimensions. For instance in NWP models localisation is done horizontally and vertically differently. How would this work under your method?

- The regularisation parameter of the Bregman divergence function is crucial in the process described here. As I understand, it can be a scalar or matrix. In fact you have used a scalar for L96 and a matrix for SWE and I guess this depends on the 'nature' of the state variables. What would the process to find either an optimal or a useful lambda be in a large model? I am just asking to discuss this a little further, not to do it. Also, can one obtain a physical interpretation of the parameter. With localisation one can speak, for instance, of de-correlation length-scales.

6. PLOS authors have the option to publish the peer review history of their article (what does this mean?). If published, this will include your full peer review and any attached files.

Reviewer #1: **Yes: **Peter Jan van Leeuwen

Reviewer #2: **Yes: **Elias D. Nino-Ruiz

Reviewer #3: **Yes: **Javier Amezcua

---

## [Author Response · Author response to Decision Letter 0]

28 Dec 2020

Please see attached pdf for the response to the reviewers. We also attached a letter for the editor.

---

## [Decision Letter · Decision Letter 1]

25 Jan 2021

PONE-D-20-14752R1

Penalized Ensemble Kalman Filters for High Dimensional Non-linear Systems

PLOS ONE

Dear Dr. Hou,

Thank you for submitting your manuscript to PLOS ONE. Both reviewers and me found that you successfully addressed the previous concerns. The paper requires some minor revision before it could be considered suitable for publication, and we invite you to submit a revised version of the manuscript that addresses the few remaining points identified by Reviewer #1.

We look forward to receiving your revised manuscript.

Kind regards,

Andrea Storto

Academic Editor

PLOS ONE

Reviewers' comments:

Reviewer's Responses to Questions

**Comments to the Author**

1. If the authors have adequately addressed your comments raised in a previous round of review and you feel that this manuscript is now acceptable for publication, you may indicate that here to bypass the “Comments to the Author” section, enter your conflict of interest statement in the “Confidential to Editor” section, and submit your "Accept" recommendation.

Reviewer #1: (No Response)

Reviewer #3: All comments have been addressed

2. Is the manuscript technically sound, and do the data support the conclusions?

Reviewer #1: Yes

Reviewer #3: Yes

3. Has the statistical analysis been performed appropriately and rigorously? 

Reviewer #1: Yes

Reviewer #3: Yes

4. Have the authors made all data underlying the findings in their manuscript fully available?

Reviewer #1: Yes

Reviewer #3: Yes

5. Is the manuscript presented in an intelligible fashion and written in standard English?

Reviewer #1: Yes

Reviewer #3: Yes

6. Review Comments to the Author

Reviewer #1: The authors did a very good job in replying to the issues raised, and I suggest acceptance after the following minor comments have been taken into consideration:

Line 200: number of ensemble members

Line 203: the order of ... Please fix notation.

Line 408: There is only one ensemble.

Second reference: The second author is P.J. Van Leeuwen

Algorithm 1: As it stands now this is just a sloppy code for an EnKF. The important steps that set this method apart from an EnKF are missing. Please provide a pseudo code for the whole algorithm. Also note that the output is an ensemble, not just the ensemble mean. Finally, A,Q, R and D need to be defined too.

Algorithm 2: I suggest removing this algorithm as an EnKF is never coded up this way for high-dimensional systems. Instead, the solution is found in ensemble space.

Reviewer #3: All my comments have been addressed in an appropriate way. I also read over to the comments from other reviewers and I think they have addressed too.

7. PLOS authors have the option to publish the peer review history of their article (what does this mean?). If published, this will include your full peer review and any attached files.

Reviewer #1: No

Reviewer #3: **Yes: **Javier Amezcua

---

## [Author Response · Author response to Decision Letter 1]

16 Feb 2021

Response to Reviewers is in the attached file

---

## [Editor Report · Decision Letter 2]

19 Feb 2021

Penalized Ensemble Kalman Filters for High Dimensional Non-linear Systems

PONE-D-20-14752R2

Dear Dr. Hou,

We’re pleased to inform you that your manuscript has been judged scientifically suitable for publication and will be formally accepted for publication once it meets all outstanding technical requirements.

Kind regards,

Andrea Storto

Academic Editor

PLOS ONE
---

## [Editor Report · Acceptance letter]

3 Mar 2021

PONE-D-20-14752R2

Penalized Ensemble Kalman Filters for High Dimensional Non-linear Systems

Dear Dr. Hou:

I'm pleased to inform you that your manuscript has been deemed suitable for publication in PLOS ONE. Congratulations! Your manuscript is now with our production department.

Kind regards,

on behalf of

Dr. Andrea Storto

Academic Editor

PLOS ONE